# The Sustainable Development Goals for Education and Research in the Ranking of Green Universities of Mahasarakham University

Woraluck Sribanasarn [1], Rapeepat Techarungruengsakul [2], Mathinee Khotdee [3], Sattawat Thuangchon [2], Ratsuda Ngamsert [4], Anujit Phumiphan [5], Ounla Sivanpheng [6] and Anongrit Kangrang [2,*]

1   Division of Building and Grounds, Mahasarakham University, Kantharawichai District, Maha Sarakham 44150, Thailand; woralak.s@msu.ac.th
2   Faculty of Engineering, Mahasarakham University, Kantharawichai District, Maha Sarakham 44150, Thailand; rapeepat.tec@msu.ac.th (R.T.); sattawat.t@msu.ac.th (S.T.)
3   Faculty of Architecture, Urban Design & Creative Arts, Mahasarakham University, Kantharawichai District, Maha Sarakham 44150, Thailand; mathinee.k@msu.ac.th
4   University Industry Cooperation Center, Mahasarakham University, Kantharawichai District, Maha Sarakham 44150, Thailand; ratsuda.n@msu.ac.th
5   School of Engineering, University of Phayao, Phayao District, Phayao 56000, Thailand; anujit.ph@up.ac.th
6   Faculty of Water Resources, National University of Laos, Vientiane 01020, Laos; o.sivanpheng@nuol.edu.la
*   Correspondence: anongrit.k@msu.ac.th

**Abstract:** This research aims to review the educational and research operations of Mahasarakham University and propose development directions for the university to meet the sustainable development criteria for ranking as a green university. This involves gathering data and analyzing the results of operations over the past 3 years, then using this analysis to synthesize the lessons learned and develop guidelines for sustainable development in the coming years. The study utilizes a robust methodology involving policy analysis, strategic planning, performance evaluation, and data integration. The study found that the university's policies and strategies implemented following the annual performance evaluation criteria align with the green university assessment criteria for education and research. This has resulted in minor improvements in the curriculum for sustainability. However, community service projects for sustainability have increased by 89.10%, while funding for sustainable research and academic publications related to sustainability has decreased. In summary, the projects under the university's development strategy adequately support and drive suitable development activities. Nevertheless, the university must continuously review its operations to adapt to changing contexts, budget constraints, evolving competition, and long-term development towards the implementation of the sustainable development goals (SDGs) to develop in the future.

**Keywords:** green university; sustainability; education and research; Mahasarakham University; UI Green Metric

## 1. Introduction

Universitas Indonesia (UI) initiated world university rankings in 2010, later known as the UI Green Metric World University Rankings, to measure campus sustainability efforts. It was intended to create an online survey to portray sustainability policies and programs for universities around the world [1–4].

The UI Green Metric conceptual framework of environment, economy, and equity served as a general foundation for our rankings at Mahasarakham University. The rating criteria and indicators are meant to be universally applicable. The indicators and weightings were created with as little bias as possible. Data collection and submission are simple tasks that take a modest amount of staff time to complete. The 2010 edition of the UI Green Metric included 95 institutions from 35 different countries, including 18 from the United States,

35 from Europe, 40 from Asia, and 2 from Australia. Last year, there were 1183 colleges from 85 different nations that took part in 2023 [5–10]. This demonstrates that the UI Green Metric is the first and only university rankings system in the world that considers sustainability, as shown in Figure 1.

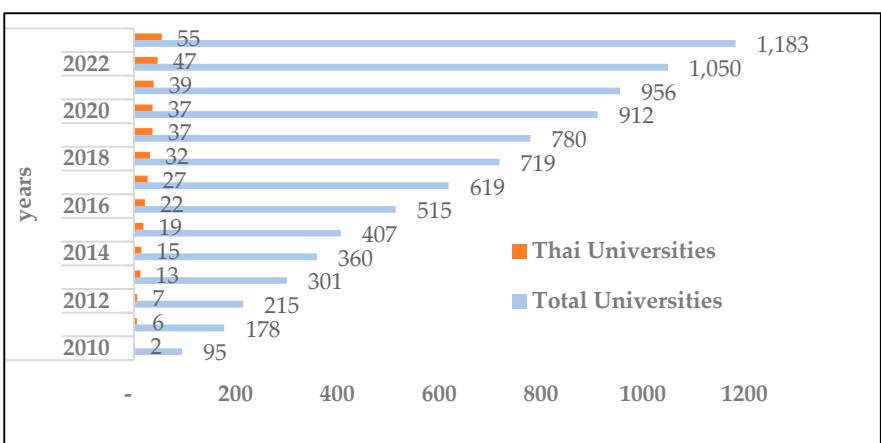

**Figure 1.** Number of universities participating in the ranking from 2010–2023.

The rankings are intended to support scholarly discussions on campus greening and sustainability in education. They encourage social change spearheaded by universities with an eye on sustainability aims and serve as a tool for higher education institutions (HEIs) all over the world to analyze their own on-campus sustainability. Governments, local and international environmental organizations, and the general public should be made aware of campus sustainability initiatives. The key points in 2023 were "Innovation, Impacts, and Future Direction of Sustainable Universities". We would like to focus on universities' efforts to continue their sustainability programs and policies, innovation, impacts, and future directions to become sustainable universities based on the UI Green Metrics and SDGs [11–16].

The green university ranking initiative began in 2010, with a total score of 10,000 points divided into six components: Setting and Infrastructure (SI), Energy and Climate Change (EC), Waste Management (WS), Water Usage (WR), Transportation (TR), and Education and Research (ED). Education and research are particularly crucial dimensions, as they significantly drive the advancement of universities in Thailand, aligning with the criteria for assessing the quality assurance of education in the country [17–27].

Mahasarakham University is a state university with missions in education management, research, academic services, and cultural preservation. Currently, it offers education at all higher levels, including undergraduate, graduate, and doctoral programs. The university continuously develops new sustainable undergraduate programs and supports research funding to enhance its ability to create new innovations that benefit both the community and society. Mahasarakham University drives its education and research operations continuously under sustainable principles, with a total of 55 sustainable programs out of a total of 95 programs [28,29].

Mahasarakham University has been participating in the Green University Ranking since 2011. Education and research data are crucial for assessing the quality of education according to Thai standards and can also be used for the UI Green Metric World University Ranking. Therefore, Mahasarakham University's strengths lie in its adherence to Thai national curriculum standards and its alignment with the criteria for Green University Ranking assessments in education and research. The university's sustainable education and research operations aim to develop the university, with the vision of becoming a leading university in Asia. This vision is pursued through missions and strategic plans that align with sustainable education and research initiatives. These initiatives include producing graduates aligned with global societal needs; conducting research and innovation to achieve international excellence; providing academic services to meet the demands of industry and

society; preserving and promoting Isaan culture internationally; and managing personnel for university excellence. These five strategic thrusts drive the university's efforts to achieve the set goals for each indicator in education and research according to the Green University Ranking criteria [30–32].

To achieve sustainable development, Mahasarakham University must conduct budget analyses and assess its activities, projects, and outcomes in education and research each year. This process enables the university to review and improve operations in the following year in line with the criteria and indicators of the Green University Ranking assessment. It allows for accurate adjustments according to the standards, resulting in improved assessments of education and research activities. Additionally, analyzing past operational activities each year helps to extract the valuable lessons learned, leading to effective adjustments in the university's development strategies, as outlined in its objectives.

Consequently, this study endeavors to gather and scrutinize retrospective data spanning the last three years to extract the valuable insights and discern lessons learned. These findings are intended to inform strategic initiatives aimed at propelling Mahasarakham University towards sustainable development, aligning with the benchmarks outlined in the Green University Ranking. By doing so, this research seeks to provide a blueprint not only for Mahasarakham University, but also as a model for other institutions, facilitating their journey towards sustainable advancement.

## 2. Materials and Methods

### 2.1. Research Area

Mahasarakham University (MSU), Khamriang Campus, was set up in Kantarawichai District, approximately seven kilometers from the original campus, with 17 faculties, 2 colleges, and 1 school currently operating. MSU has been widely recognized as one of Thailand's fastest-growing universities. Its total enrollment has also increased from fewer than 10,000 in its earlier years to more than 40,000 students at present. Many faculty buildings have been constructed on the Khamriang campus, which is now the administrative and academic center of the university. The total area of the main campus (Khamriang) is 1,697,600 m$^2$, as shown in Figures 2 and 3.

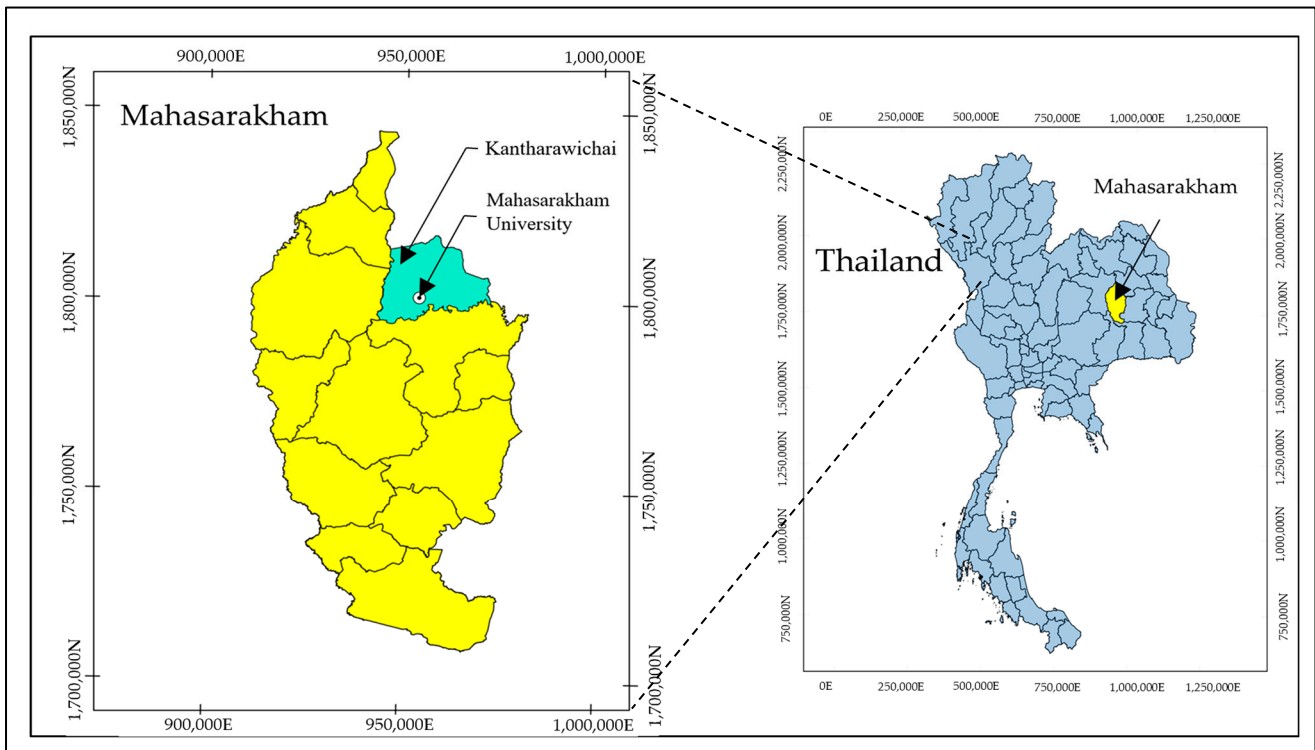

**Figure 2.** Mahasarakham Province.

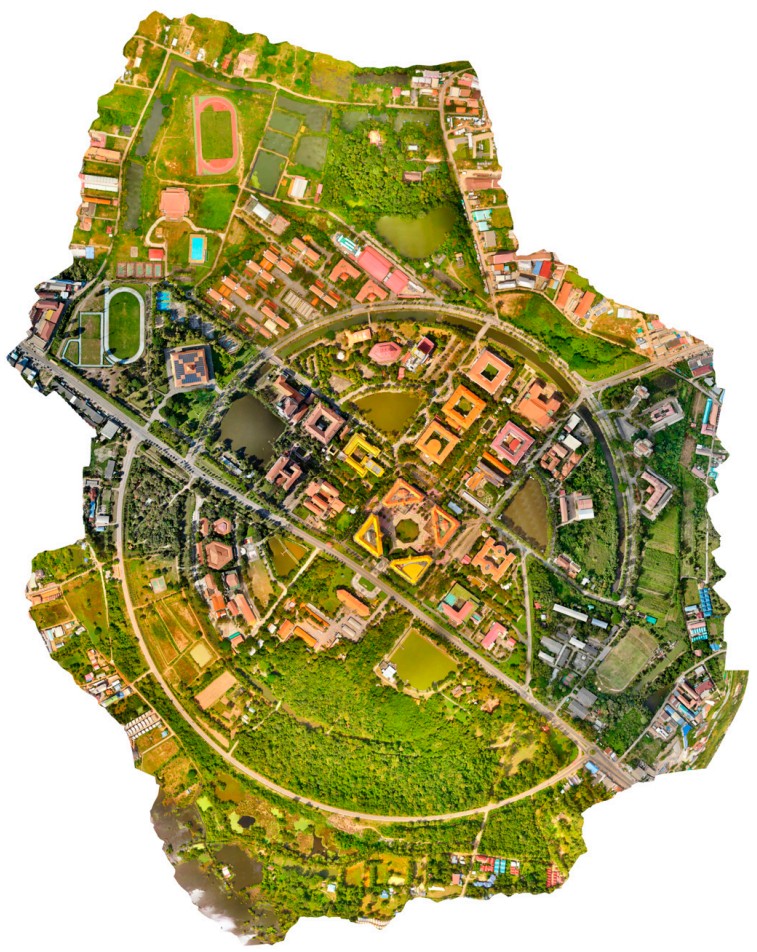

**Figure 3.** Mahasarakham University.

*2.2. Methodology*

2.2.1. The Criteria

In 2023, the categories and the weighting of points were modified to accommodate new questions and are shown in Tables 1 and 2. The assessment of each indicator will exhibit variability, as delineated in Table 2, culminating in an aggregate score of 1800 points. To attain maximal scores for individual indicators, the methodology entails computing either the proportional representation of the indicator within the specified timeframe or its absolute enumeration. The achievement of full marks is contingent upon the successful management of a substantial proportion or voluminous quantity thereof.

**Table 1.** Categories used in the rankings and their weighting.

| No. | Categories | Percentage of Total Points (%) |
|:---:|:---:|:---:|
| 1 | Setting and Infrastructure (SI) | 15 |
| 2 | Energy and Climate Change (EC) | 21 |
| 3 | Waste (WS) | 18 |
| 4 | Water (WR) | 10 |
| 5 | Transportation (TR) | 18 |
| 6 | Education and Research (ED) | 18 |
| | Total | 100 |

**Table 2.** Indicators and categories suggested for use in the 2023 rankings.

| No | Criteria | Points |
|---|---|---|
| ED1 | The ratio of sustainability courses to total courses/subjects | 300 |
| ED2 | The ratio of sustainability research funding to total research funding | 200 |
| ED3 | Number of scholarly publications on sustainability | 200 |
| ED4 | Number of events related to sustainability | 200 |
| ED5 | Number of student organizations related to sustainability | 200 |
| ED6 | University-run sustainability website | 200 |
| ED7 | Sustainability report | 100 |
| ED8 | Number of cultural activities on campus | 100 |
| ED9 | Number of university sustainability program(s) with international collaborations | 100 |
| ED10 | Number of sustainability community services project organized and/or involving students | 100 |
| ED11 | Number of sustainability-related startups | 100 |
| | Total | 1800 |

### 2.2.2. Scoring

Scoring for each item was numeric so that our data could be processed statistically. Scores were simple counts of things or responses on a scale of some sort.

### 2.3. Educational and Research Operations

Operations according to the green university ranking criteria for education and research involve defining the steps for operating according to the Deming cycle and observing four steps: Plan–Do–Check–Act, as expanded upon below [33].

### 2.3.1. Plan

The operations of green university ranking in the areas of education and research at Mahasarakham University involve the establishment of policies and strategic plans for university development. These plans are aligned with the Sustainable Development Goals (SDGs) and key global university ranking indicators [34,35]. Furthermore, the university has devised annual operational plans that correlate with the criteria for assessing green university rankings in education and research. Following the establishment of performance indicators as per these operational plans, the university has conducted analyses to assess their alignment with the evaluation criteria in education and research. This analysis serves as a conceptual framework for guiding activities related to green university ranking. The university has set targets for each indicator based on the scoring criteria, planned statistical data collection procedures, analyzed the data according to predetermined criteria, and summarized these operational data before integrating them into the system in sequence, as illustrated in Figure 4.

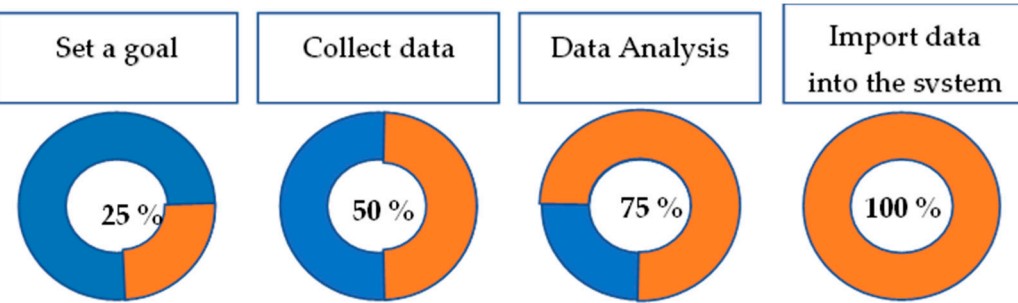

**Figure 4.** Framework procedure.

### 2.3.2. Do

Mahasarakham University actively pursues green university ranking initiatives in education and research, with the university administration, including the president and administrative faculty, playing a pivotal role in driving these efforts alongside the university's development policies. This involvement includes setting performance indicators aligned with operational plans that correlate with green university ranking activities. Internal university faculties and departments collaborate to achieve the objectives outlined in the university's operational plans. As a result of these collaborative efforts, the university has achieved success in various aspects of the green university ranking criteria, particularly in areas related to sustainable curriculum development, research funding from various sources, the dissemination of academic publications on sustainability, the organization of sustainability-related activities aligned with the assessment indicators used in green university ranking assessments conducted by student organizations and clubs, and sustainable arts and cultural projects, as well as international sustainability initiatives. The university's achievements in these areas are attributed to the cooperation of faculty, staff, and students from all faculties and departments, ensuring that their activities align with the education and research evaluation criteria and achieve the specified goals.

### 2.3.3. Check

Each year, the performance of education and research initiatives is evaluated through meetings to review and assess against set objectives. This process involves analyzing the progress made toward the predetermined goals outlined in the continuous operational plans. The oversight and monitoring of green university ranking efforts are carried out by the Green University Development and Environmental Conservation Committee, which comprises university administrators and representatives from various departments involved in overseeing the progress toward these set goals. Additionally, Mahasarakham University tracks and drives green university ranking activities through the Buildings and Facilities Management Office and the Green University Ranking Operations Committee. These efforts ensure that the reporting on green university ranking outcomes in education and research adheres to the established criteria, ensuring accuracy, completeness, and the coverage of all indicators in reporting and data entry into the system.

### 2.3.4. Action

Adjustments to operational procedures following evaluations are determined by the Green University Development and Environmental Conservation Committee. This committee evaluates the performance and green university ranking outcomes in education and research from the previous year. They analyze the progress made and identify areas where objectives have not been met. Additionally, the committee responsible for education and research operations annually reviews and studies the evaluation criteria for each indicator according to the Green University Ranking Assessment Handbook. This ensures that reporting is accurate and comprehensive, meeting the evaluation criteria effectively.

### *2.4. Data Preparation for Submission to the System*
### 2.4.1. Data Preparation

In preparation for the green university rankings in education and research, data collection efforts for the year 2023 were undertaken. This involved gathering information on the courses offered within the university, research funding to support the university's development, and activities related to sustainability. Personnel at all levels within faculties and departments contributed to driving these efforts forward. Additionally, student involvement was solicited, particularly from those engaged in research related to green university rankings in education and research. They participated in analyzing data for each indicator, compiling reports on relevant activities related to green university rankings in education and research, and collectively analyzing the progress made to ensure alignment

with the criteria for green university rankings in education and research. This collaborative effort aimed to achieve the specified goals effectively.

### 2.4.2. Data Submission System Integration

Mahasarakham University successfully entered its green university ranking data into the ranking system by the designated deadline of 31 October 2023. This involved recording operational data along with supporting information used as references for the performance indicators in each aspect, in accordance with the green university ranking criteria (Figure 5) [36].

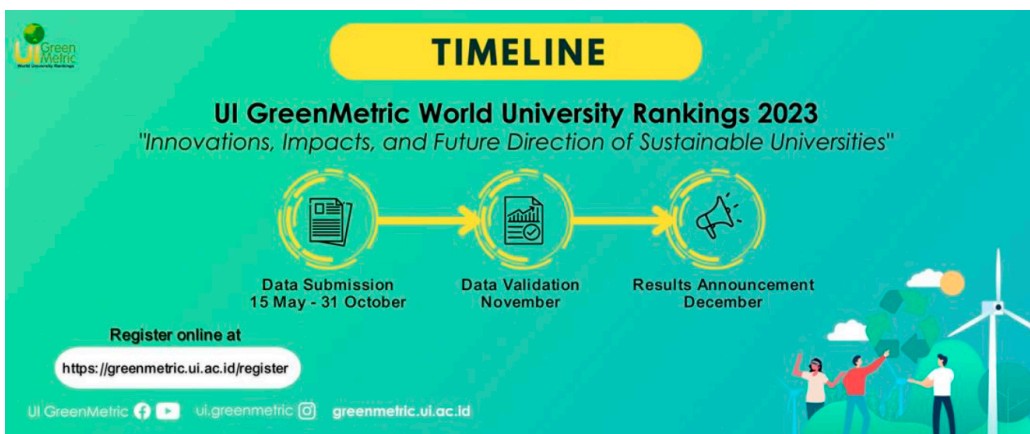

**Figure 5.** Timeline UI Green Metric World University Rankings 2023.

### 3. Results and Discussion

Since 2011, Mahasarakham University has actively participated in university rankings and has consistently achieved scores according to the green university ranking criteria. In 2015, the UI Green Metric introduced revised its evaluation criteria, categorizing them into six aspects. Mahasarakham University's performance in education and research, as per the green university ranking criteria, aligned with the university's sustainable development strategies. Performance indicators were established based on the university's annual operational plans and activities were carried out according to these plans and allocated budgets to achieve set objectives, as outlined in Table 3.

**Table 3.** Key Performance Indicators (partial) from the operational plan of Mahasarakham University aligned with sustainability initiatives.

| Strategies | Indicators | Results/Year | | | Activity Budget for 2023 |
|---|---|---|---|---|---|
| | | 2021 | 2022 | 2023 | |
| 1: Produce Graduates Meeting Global Demands | 1.1 Number of New Paradigm Courses or Sandbox Courses or International Collaboration Courses | - | - | 13 | 10 million THB |
| | 1.2 Number of International Courses | 7 | 7 | 7 | |
| 2: Foster Research and Innovation for International Excellence | 2.1 Amount of Funding from External Sources | 97.92 | 697 | 130 | 32 million THB |
| | 2.2 Number of Research Projects Funded by Foreign Sources (million THB) | 17 | 12 | 34 | |
| | 2.3 Number of Research Outputs Published in International Databases (Scopus) | 513 | 659 | 668 | |
| | 2.4 Number of Innovations or Research Outputs Derived from Research | 21 | 20 | 41 | |
| | 2.5 Number of Internationally Collaborative Articles | 178 | 96 | 126 | |

**Table 3.** *Cont.*

| Strategies | Indicators | Results/Year | | | Activity Budget for 2023 |
| --- | --- | --- | --- | --- | --- |
| | | 2021 | 2022 | 2023 | |
| 3: Provide Academic Services to Meet Industry and Community Needs | 3.1 Number of Research, Innovation, or Technology Projects Transferred to Industry/Society | 15 | 27 | 20 | |
| | 3.2 Number of Communities and Societies Served with Academic Services Resulting in Strengthening and Self-reliance | 44 | 6 | 71 | |
| | 3.3 Number of Research, Innovation, or Technology Projects Transferred to Communities and Societies through Academic Services | 44 | 6 | 55 | 1.2 million THB |
| 4: Enhance Isaan Cultural Excellence at the International Level | 4.1 Number of Internationally Outstanding Cultural Works Created | 5 | 7 | 11 | |
| | 4.2 Number of Collaborative Cultural Work Activities in the Mekong River Basin and China | 4 | 6 | 7 | |
| | 4.3 Number of Cultural Works Adding Value/Significance | 5 | 7 | 10 | 1.6 million THB |
| 5: Manage the Organization for University Excellence | 5.1 Number of Digital Technology Innovations | - | 1 | 1 | |
| | 5.2 UI Green Metric Assessment Scores | 7575 | 8200 | 8335 | 3.2 million THB |

The results of Mahasarakham University's efforts in education and research, as per the green university ranking criteria, are presented as follows.

1. Total Number of Sustainable Curriculum Offerings

Mahasarakham University has been driving operations in line with its annual operational plans, guided by the university's strategies, with a particular focus on producing graduates to meet global societal needs. This aligns with the green university ranking criterion No. 1, which measures the proportion of sustainable curriculum offerings to the total number of programs. The performance outcomes over the past three years are illustrated in Figure 6.

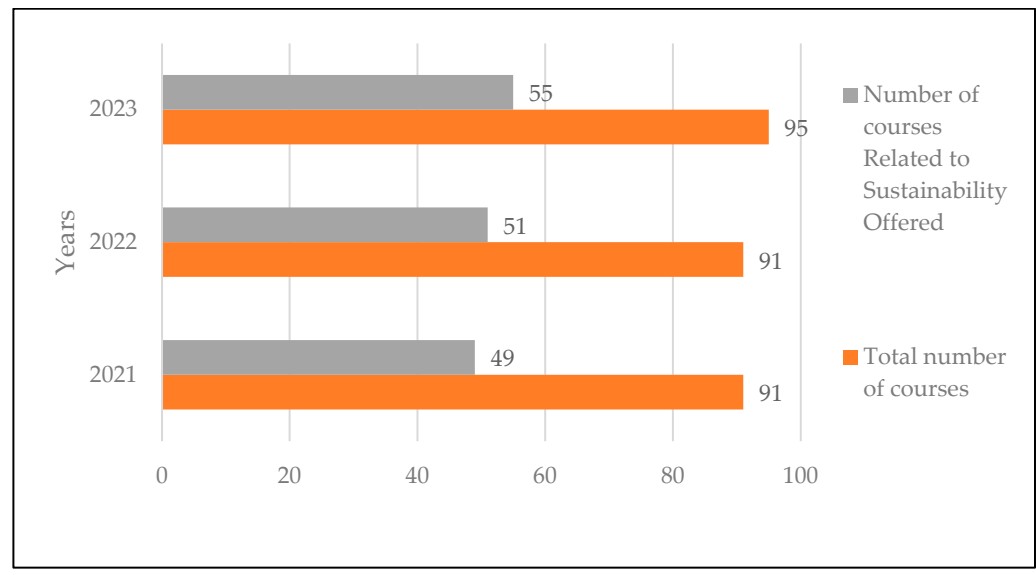

**Figure 6.** Total number of courses with sustainability embedded for courses running in 2021–2023.

From Figure 6, it is evident that Mahasarakham University has had sustainable curriculum offerings in the past and continues to develop and add new programs consistently.

The university has been actively developing and refining its curriculum offerings based on quality assurance criteria, particularly during the period from 2021 to 2023. There has been a continuous increase in the number of new and sustainable programs each year. In 2021, the university had a total of 91 programs, with 49 being sustainable. By 2023, the total number of programs had increased to 95, with the highest number of sustainable programs being 55. This increase is attributed to the university's efforts in curriculum improvement to align with its annual operational plans. The addition of new programs, sandbox programs, and international collaborations has strengthened the university's ability to offer internationally competitive education and teaching management.

Compared with other universities, it is found that Kasetsart University, which offers teaching management in subjects similar to Mahasarakham University, has a total of 99 programs, with 68 being sustainable. Kasetsart University has set its operational plan for 2023 under Strategy 1: Creative Earth Sciences for Sustainable Development of the Country. As a result of collaboration from alumni, entrepreneurs, government agencies, and the private sector, 21 new programs were introduced in 2023, reflecting an increased budget allocation for curriculum development. Kasetsart University allocated approximately THB 279.8 million for the development of new programs. In contrast, Mahasarakham University has a more limited budget allocation for the development of new and sustainable programs, which is lower than that of Kasetsart University.

2.  Research funding per Sustainable Research Fund

Mahasarakham University has established strategic priorities to foster research and innovation for international excellence. The university aims to develop the potential of researchers and the research infrastructure within the organization to increase the drive towards becoming a global leading research university. This is pursued by advancing key performance indicators according to the annual operational plan, which aligns with the criteria for assessing sustainable research funding in the Green University assessment. The research funding includes both internal university funds that support researchers in developing and advancing research, as well as external funding support. The performance over the past 3 years is illustrated in Figure 7.

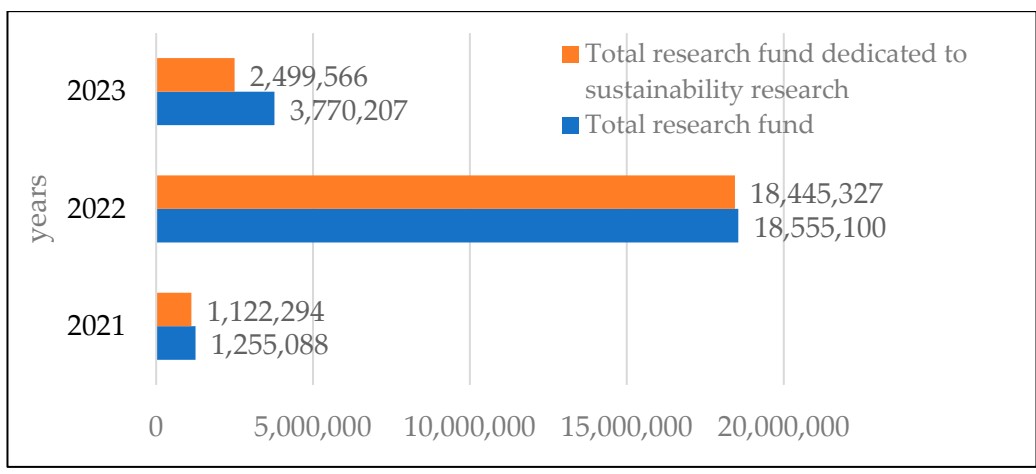

**Figure 7.** The ratio of sustainability research funding to total research funding in 2021–2023.

From Figure 7, the proportion of sustainable research funding to total research funding from 2021 to the present is shown, including support from both internal university funds and external sources. This continuous support has facilitated the ongoing development of research and innovation [37,38]. As depicted in the figure, Mahasarakham University received its highest research funding in 2022, totaling THB 18,555,100, primarily from the Thai government to revitalize the grassroots economy and communities in the agricultural sector, including state-owned enterprises. Conversely, in 2021, the university received its lowest research support, amounting to THB 1,255,088, largely due to the impact of the

COVID-19 situation. These fluctuations in research funding reflect the adjustments made in response to the changing circumstances brought about by the COVID-19 pandemic.

When compared to other universities, it was found that Kasetsart University, which operates in a similar capacity to Mahasarakham University, received research funding of THB 88,280,000 in 2023. Additionally, it set a target of producing 300 research outputs in the same year. Kasetsart University is among the institutions that received the highest government funding in 2023, totaling THB 5,077,400,000, with THB 829,900,000 allocated specifically for research support. This amount is significantly higher than the funding received by Mahasarakham University from the government (approximately five times higher).

3.    Number of sustainable publications

Mahasarakham University sets performance indicators according to its annual operational plan, aiming to generate high-quality research and innovations that are internationally recognized, thereby propelling it towards becoming a leading research university on a global scale. The outcomes of its efforts are reflected in the number of sustainable research publications related to sustainability produced over the past three years, as shown in Table 4.

**Table 4.** Number of academic publications related to sustainability from 2021–2023.

| Number of Publications | 2021 | 2022 | 2023 |
|---|---|---|---|
| Publication related to "Green and Sustainability" | 262 | 309 | 298 |
| Publication related to "Environment" | 1160 | 1190 | 1220 |
| Publication related to "Renewable Energy" | 775 | 847 | 801 |
| Publication related to "Climate Change" | 923 | 1050 | 958 |
| Total number of publications | 3120 | 3396 | 3277 |

Table 4 illustrates the number of academic research publications related to sustainability produced from 2021 to 2023 on the Google Scholar website, showing a consistent increase in the number of publications, in alignment with the university's annual operational plan. In pursuit of Strategy 2, to create research and innovation excellence at the international level, Mahasarakham University has seen a continuous rise in the number of publications related to sustainability. The highest number of academic research publications related to sustainability was recorded in 2022, with 3396 publications, while the lowest was in 2021, with 3120 publications. This trend reflects the adjustments in the budget allocation for research and innovation activities due to the impact of the COVID-19 pandemic, resulting in a reduction in operational budgets [39,40].

When compared with other universities, Mahidol University had the highest number of academic research publications in 2023. They had 1240 publications related to 'Green and Sustainability', 3810 publications related to 'Environment', 2950 publications related to 'Renewable Energy', and 3650 publications related to 'Climate Change', totaling 11,650 publications related to sustainability in 2023. This number of academic research publications is significantly higher than that of Mahasarakham University, mainly because Mahidol University boasts a superior research environment. It receives substantial government funding, ranks first in the country, and has a larger research faculty and access to advanced research equipment and facilities.

4.    Number of sustainable community service projects organized and/or involving students

Mahasarakham University has set strategic priorities in its academic services to meet the needs of both the industry and the community. These priorities are measured according to operational plans aligned with the criteria for assessing green universities in education and research. One of the initiatives is community service for sustainability, which involves organizing and/or engaging students in community service projects. The results of these initiatives over the past three years are illustrated in Table 5.

**Table 5.** Number of sustainable community service projects organized and/or involving students.

| Projects | 2021 | 2022 | 2023 |
|---|---|---|---|
| Sustainable Community Service Projects | 44 | 6 | 71 |

Table 5 illustrates the number of community service projects organized and/or involving students from 2021 to 2023, aligning with the university's operational plans in Strategy 3 regarding academic services to meet the needs of both industry and society. From Table 5, it can be observed that Mahasarakham University has seen an increase in its number of community service projects. In 2022, the maximum number of projects was 6, while in 2023, the maximum number reached 71. This increase reflects the university's increased budgetary support for academic service initiatives.

When compared to other universities, Kasetsart University operates in a manner similar to Mahasarakham University. Kasetsart University sets goals according to its operational plan. In its Strategy 1.3, under the overarching strategy of fostering and supporting community and social development, it aims to involve its staff and students in community development. One of its objectives is to undertake a certain number of projects or initiatives aimed at improving various aspects of community life, such as quality of life, community economy, community learning centers, and environmental conservation. For instance, the University to Tambon (U2T) project targets 20 sub-districts, with 243 projects being implemented (Kasetsart University Operational Plan for Fiscal Year 2023, Kasetsart University, 2023).

**4. Conclusions**

Mahasarakham University has set goals for developing its efforts in the university green rankings. Key factors contributing to achieving its set goals include the sustainable university policy established by university management. This policy provides a framework for operational plans, sustainable budget allocations for university development, project activities, and the collaboration of staff and students through various committees. For example, committees responsible for university green ranking initiatives and evaluation criteria invite students to participate in research, collaborate with student organizations for sustainability projects, and conduct various advocacy campaigns. Quality assurance processes are necessary during project implementation to ensure clear planning, goal setting, the monitoring of key performance indicators, and the evaluation of success or failure. This allows for adjustments in future operations to ensure the continuous development and achievement of goals across all aspects of university green ranking initiatives.

The continuous development of efforts in education and research for the university green rankings is the result of the university's sustainability policy. This policy encompasses activities related to the university's green initiatives and is linked to the university's sustainability-oriented strategic plans. There are 55 courses related to sustainability out of a total of 95 courses offered by the university. Adequate budget allocations for sustainability-related activities further contribute to Mahasarakham University's strengths in research and innovation for sustainability, with 55 research outputs achieved. Additionally, the establishment of research centers fosters collaboration with communities and provides learning opportunities for students. These outcomes stem from the research endeavors of university staff, leading to success in education and research efforts for sustainability.

In addition, Mahasarakham University has continuously developed and improved its operations to meet the criteria for the green university ranking assessment. This involves analyzing performance results from previous years, studying criteria where objectives have not yet been met, and planning adjustments to enhance data collection, analysis, and verification processes. This includes scrutinizing data from each committee responsible for the green ranking assessment criteria and ensuring accurate and complete data integration into the system, as per the established standards. These efforts aim to achieve the set objectives and foster genuine sustainable development.

**Author Contributions:** Conceptualization, W.S., R.T., M.K., S.T., R.N., A.P., O.S. and A.K.; methodology, W.S., R.T., M.K., S.T., R.N., A.P., O.S. and A.K.; validation, A.P. and A.K.; formal analysis, W.S., A.P. and A.K.; investigation W.S. and A.K.; writing—original draft preparation, W.S., A.P. and A.K.; supervision, A.P. and A.K.; and writing—review and editing, W.S., A.P. and A.K. All authors have read and agreed to the published version of the manuscript.

**Funding:** This research was financially supported by Mahasarakham University.

**Institutional Review Board Statement:** Not applicable.

**Informed Consent Statement:** Not applicable.

**Data Availability Statement:** This study did not report any data.

**Conflicts of Interest:** The authors declare no conflicts of interest.

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
