# Peer review of "The Sustainable Development Goals for Education and Research in the Ranking of Green Universities of Mahasarakham University"

_sustainability, doi:10.3390/su16093618_

Round 1

Reviewer 1 Report

Comments and Suggestions for Authors

Dear authors,

I find this paper quite engaging, as it presents a current and interesting perspective on exploring the educational efforts and research operations that Mahasarakham University undertakes to promote sustainability.

The university’s endeavors are well-articulated. The authors delve into past literature to establish a conceptual framework for the key themes explored in the paper—namely, education and research, university curricula, and sustainability. While the topic is compelling for the journal's audience, I believe the following aspects need improvement.

1. The literature review underpinning this analysis's methodology is comprehensive. When discussing the initiative to align education and research initiatives to current sustainability trends or requirements, exploring the challenges posed by current regulatory changes mandating sustainability expertise quantification, as well as the relationships of green intellectual capital and sustainability expertise, could enhance the depth of your discussion. By broadening the research framework informing your topic, you can highlight the novelty this manuscript brings to literature. Here are a few examples that could contribute to your discussion: https://doi.org/10.3390/su151813913, https://doi.org/10.1108/BIJ-11-2021-0641.

2. Please revise the description of the Scoring used in the analysis. The current description in lines 132-133 is not clear.

Good luck with your future research!

Author Response

We really appreciate the reviewers' comments, which are very detailed and very helpful in improving our manuscript. We have made a major revision to our manuscript. We have improved our manuscript following reviewers' comments for round 1. All of the changes have been modified in the revised manuscript. We have addressed all of the comments which are shown below for each one, in which the reviewer’s comments are in black text and authors’ responses are in green text. Please find the attached file of response to reviewers.

Reviewer 2 Report

Comments and Suggestions for Authors

After a thorough review of the manuscript titled "Sustainable Development Goals for Education and Research in the Ranking of Green Universities of Mahasarakham University" by Woraluck Sribanasarn et al., here are the detailed reviewer comments for the authors:

Reviewer Comments to Author

  1. Title and Abstract:

    • The abstract concisely outlines the research objectives, methodology, and key findings. However, it could benefit from a brief mention of the significance of the findings and their implications for the field of sustainable education.
  2. Introduction:

    • The introduction provides a good background on the UI Green Metric World University Rankings and the importance of sustainability in universities. It would be enhanced by including a more explicit statement of the research question and objectives towards the end of the section.
  3. Methodology:

    • The methodology section clearly describes the research area and the data collection process. Including more detail about the analytical techniques used to synthesize the lessons learned and develop guidelines would improve the transparency and replicability of the study.
  4. Results and Discussion:

    • The results are presented in a clear and structured manner, with a good integration of tables and figures. The discussion could be expanded to compare these findings with those of other universities not just in Thailand but globally, to provide a broader context.
  5. Conclusions:

    • The conclusion summarizes the key findings effectively. It could be strengthened by clearly stating the practical implications of the research for university administrators and policymakers involved in sustainability and green university initiatives.
  6. Figures and Tables:

    • The figures and tables are informative but some (e.g., Figure 1, Table 3) could benefit from more detailed captions explaining what the data shows and its relevance to the study's objectives.
  7. References:

    • The reference list appears comprehensive but should be checked for the most recent and relevant sources, particularly those published in the last 2-3 years, to ensure the manuscript reflects the current state of research.
  8. Language and Grammar:

    • The manuscript is generally well-written. Minor grammatical and typographical errors should be corrected for clarity. Professional proofreading is recommended to polish the language further.
  9. Data Availability Statement:

    • The manuscript states, "This study did not report any data." Given the empirical nature of the research, clarifying this statement is crucial. If applicable, details on how the data can be accessed or the reasons for any restrictions should be provided.
  10. Ethical Considerations:

    • The manuscript does not mention ethical considerations or approvals, which might be relevant given the data collection from an educational institution. The authors should clarify whether any ethical approvals were sought or why they might not be necessary.
Comments on the Quality of English Language

Minor overall improvements. 

Author Response

(The authors gave the same response as above.)

Round 2

Reviewer 2 Report

Comments and Suggestions for Authors

No further comments. All my previous points were addressed professionally by the authors.